

# Clinical outcomes in lupus nephritis patients treated with belimumab in real-life setting: a retrospective comparative study in China

Zishan Lin[1,2,3,*], Bingjing Jiang[1,2,3,*], Wenfeng Wang[1,2,3], Caiming Chen[1,2,3], Yujia Wang[1,2,3], Jianxin Wan[1,2,3] and Yanfang Xu[1,2,3]

[1] Department of Nephrology, Blood Purification Research Center, the First Affiliated Hospital, Fujian Medical University, Fuzhou, China
[2] Research Center for Metabolic Chronic Kidney Disease, the First Affiliated Hospital, Fujian Medical University, Fuzhou, China
[3] Department of Nephrology, National Regional Medical Center, Binhai Campus of the First Affiliated Hospital, Fujian Medical University, Fuzhou, China
[*] These authors contributed equally to this work.

## ABSTRACT

**Objective**. The use of belimumab in treating lupus nephritis (LN) patients in China is still in its early stages. This retrospective comparative study aims to delineate the disease activity, associated therapies, clinical outcomes, and adverse events among LN patients treated with belimumab, reflecting real-world experience in southeastern China.

**Methods**. From May 2020 to December 2023, 54 LN patients treated with belimumab and 42 LN patients treated with conventional therapy were enrolled. All patients had a follow-up period of more than 3 months. The general information, presenting clinical and laboratory data, and outcomes were collected and compared.

**Results**. At 3 months of belimumab treatment, compared to baseline, there was a decrease in proteinuria from 74.1% to 64.8% ($p < 0.001$), a reduction in hematuria from 59.3% to 37.0% ($p = 0.008$), and an increase in partial or complete renal response from 53.7% to 75.9% ($p < 0.001$). The median SLEDAI score decreased from 10 to 5 ($p < 0.001$), and the proportion of patients achieving low lupus disease activity state (LLDAS) increased from 11.11% to 16.67% ($p < 0.001$) by the 3-month evaluation. Notably, there were significant reductions in oral corticosteroid dosages, with a median decrease from 30 to 17.5 mg/day ($p < 0.001$) by 3 months, and the proportion of patients requiring >5 mg/day of steroids decreased from 88.89% at baseline to 79.07% at six months ($p < 0.001$). Compared to the conventional therapy group, the belimumab group experienced a significant reduction in median steroid dosage and increased the proportion of patients achieving remission or LLDAS. The incidence of treatment-emergent adverse events (TEAEs) was significantly lower in the belimumab group (29.6% *vs* 52.4%, $p = 0.024$).

**Conclusion**. These findings support the potential of belimumab to improve renal and serological parameters, reduce disease activity, lessen corticosteroid dependence, and decrease the risk of TEAEs, demonstrating its safety and efficacy as an adjunct therapy in LN management.

Corresponding author
Yanfang Xu, xuyanfang99@hotmail.com

## INTRODUCTION

Systemic lupus erythematosus (SLE) is a complex autoimmune disease affecting various tissues, from the skin to internal organs (*Anders et al., 2020*). Lupus nephritis (LN), one of the most severe and common manifestations of SLE, develops in 40–60% of SLE patients in the early stages. This manifestation is notably more prevalent among Chinese patients and imposes a significant economic burden (*Feng et al., 2011*). Pathologically, LN is characterized by the deposition of immune complexes within the endothelial and subendothelial layers of the affected kidneys, resulting in extensive damage and nephron loss during the acute phase. Without effective treatment, LN can lead to irreversible structural or functional damage. Renal involvement serves as an independent risk factor for a poor prognosis in patients with SLE, associated with prolonged hospitalization and increased mortality rates (*Feng et al., 2011*; *Davidson et al., 2018*; *Tanaka et al., 2018*). The management of LN typically requires long-term steroids and immunosuppressive therapy to control disease activity and preserve kidney function. However, the remission rate among LN patients remains unsatisfactory, with approximately 10–20% progressing to end-stage renal disease (*Yu et al., 2022*). Timely disease management significantly increases the 10-year survival rate from 46% to 95%, highlighting the critical importance of early recognition and intervention in LN (*Chen et al., 2008*).

B cells, the precursors to plasma cells responsible for antibody production, play a pivotal role in the pathogenesis of SLE. In 2011, belimumab, a biologic agent, received approval for treating patients with active SLE. This recombinant human monoclonal antibody targets the soluble B-lymphocyte stimulator, a key regulator of B lymphocyte function, offering a novel approach to SLE management (*Hahn, 2013*; *Scholz et al., 2008*). The last decade has seen belimumab emerge as a significant regulator in SLE treatment, as evidenced by numerous clinical studies (*Levy et al., 2021*). Notably, a landmark two-year study involving 448 LN patients demonstrated that adjunctive belimumab treatment resulted in significantly improved primary efficacy renal responses compared to standard therapy alone (*Furie et al., 2020*). This enhancement in treatment outcomes was further supported by a recent observational study, underscoring belimumab's benefit when added to standard therapy in LN patients (*Sishi et al., 2023*). Additional real-world studies have corroborated these findings, showing that belimumab reduces disease activity, flare rates, and oral glucocorticoid dose in LN patients (*Roberts et al., 2023*; *Tan et al., 2023*). The drug's safety and efficacy have been validated in specific patient populations, including children, those undergoing dialysis, and post-transplant patients, broadening its applicability (*Tan et al., 2023*; *Liu et al., 2022*; *Binda et al., 2020*). Despite these positive outcomes, caution is warranted due to the association of belimumab with various adverse reactions, including infections, acute pancreatitis, and myositis (*Tan et al., 2023*; *Wise & Stohl, 2019*). Moreover, the safety and efficacy of belimumab in patients with an estimated glomerular filtration

rate (eGFR) less than 30 mL/min/1.73 m$^2$ remain to be fully determined, highlighting an area for further research.

Therefore, belimumab treatment emerges as a promising strategy for disease modification in LN. However, further investigation is imperative to confirm its safety and efficacy, particularly among Chinese patients with LN. This study aims to evaluate the clinical efficacy and safety of belimumab in LN patients within a real-life setting, providing crucial insights into its application and outcomes in this demographic.

## METHODS AND MATERIALS

### Patient selection

Between May 2020 and December 2023, patients at the First Affiliated Hospital of Fujian Medical University, who were undergoing combined therapy with belimumab and had a follow-up period of at least three months, were enrolled in the belimumab group. Patients who received conventional therapy during the same timeframe and underwent routine follow-up were enrolled in the conventional therapy group. At the time of enrollment, these patients in the belimumab group were administered belimumab for the first time. The diagnosis of SLE adhered to the SLICC/ACR SLE classification criteria or the revised 1997 American College of Rheumatology criteria. The confirmation of LN was based on renal biopsy results or sustained positive findings in urine analysis (*DIGO, 2021*; *Tiao et al., 2016*; *Hochberg, 1997*). Patients with less than three months of follow-up were excluded from the study.

This study was performed in accordance with the Helsinki Declaration and approved by the ethics committee of the First Affiliated Hospital of Fujian Medical University [2015]084-2. Written informed consent was obtained from each patient.

### Data collection

Clinical data were retrieved from patients' medical records upon hospitalization, encompassing the course of SLE and LN, clinical manifestations, age, gender, and the initial immunosuppressive regimen combined with belimumab. Biological data comprised serum creatinine levels, complete blood counts, complement 3 (C3), complement 4 (C4), anti-dsDNA antibodies, 24-hour proteinuria, urinary albumin/creatinine ratio (UACR), proteinuria, hematuria, leukocyturia, cylindruria, and the proportion and absolute count of CD19$^+$ B cells. SLEDAI-2K scores were recorded to evaluate clinical disease activity (*Gladman, Ibañez & Urowitz, 2002*). Renal biopsy specimens were evaluated using light microscopy, direct immunofluorescence, and electron microscopy. Information on concomitant therapeutic agents, daily glucocorticoid dosage, and adverse events (AEs) was gathered. Patients received regular follow-ups, and data corresponding to month 3, month 6, and month 12 were collected for analysis.

### Definitions

Complete renal response (CRR) was defined as proteinuria less than 0.5g/24 h or 50mg/mmol, accompanied by improvement or stabilization in renal function (at least normal or no worse than 10% below baseline). Partial renal response (PRR) was defined

as proteinuria less than 3.0g/24 h with a reduction of more than 50% from baseline, along with improvement or stabilization in renal function (*DIGO, 2021*).

Patients were considered to be in remission if they met the following criteria: clinical SLEDAI-2K score = 0 (serology excluded), PGA score <0.5, and receiving an equivalent glucocorticoid dose of ≤5 mg/d with stable antimalarials, immunosuppressive therapies, and biological agents, in accordance with the recommendations of the DORIS Task Force (*Van Vollenhoven et al., 2021*). Lupus Low Disease Activity State (LLDAS) is defined as SLEDAI-2K ≤4, inactivation of central organ system, PGA score ≤ 1.0, absence of new activity, steroids dosage ≤7.5mg/day, and standard dosages of antimalarial drugs, immunosuppressants, and biological agents (*Tselios, Gladman & Urowitz, 2019*). Based on a previous study, disease severity subgroups were categorized according to SLEDAI scores: mild (0–6), moderate (7–12), and severe (>12) (*Liu et al., 2022*).

### Belimumab therapy
Belimumab administration followed recommended guidelines (*Ward & Tektonidou, 2020*). Briefly, patients received 10 mg/kg of belimumab on day 1, 15, and 29, and then received the same dose every 28 days. Vital signs of patients were monitored during infusion using an electrocardiogram monitor.

### Statistical analysis
Statistical analysis was performed using SPSS 27.0 (SPSS, Chicago, IL, USA) and GraphPad version 8.3.0 (GraphPad Software, San Diego, CA, USA). The Shapiro–Wilk test was used to assess the normality of the data. Continuous variables with normal distribution were presented as mean ± standard deviation (SD), and the paired *t* test was used for comparison within the same group. The continuous variables with non-normal distribution were described by median with interquartile range (IQR), and the Wilcoxon Signed-Rank Test was used for comparison within the same group, while the Mann–Whitney U Test was used for comparison between groups. Categorical variables were expressed as percentages and compared using the $\chi 2$ test or Fisher exact test. A *p*-value <0.05 was considered statistically significant.

## RESULTS
### Patient characteristics and indications for initiation of belimumab
A total of 96 patients were enrolled in this study, comprising 54 in the belimumab group and 42 in the conventional therapy group. Among the individuals receiving belimumab therapy, all had used it for at least 3 months, 43 for 6 months, and 18 for 12 months. The primary objectives for initiating belimumab treatment included controlling active SLE or LN in 83.3% of cases, reducing steroid dependence in 11.1%, and preventing disease flares in 5.6%. As delineated in Table 1, in terms of follow-up duration, the belimumab group had a significantly shorter duration compared to the conventional therapy group (7 months *vs* 37 months, *p* <0.001). Demographically, the median age in the belimumab group was 37 [23.25, 45] years, with 87.0% (47/54) females. No significant differences were found between two groups.

**Table 1   Patient characteristics at baseline.**

| Characteristic | Conventional therapy (*n* = 42) | Belimumab (*n* = 54) | *P* value (Belimumab *vs* Conventional) |
|---|---|---|---|
| Female sex, *n* (%) | 38 (90.5) | 47 (87.0) | 0.840 |
| Age, years | 32 [24.25, 40.5] | 37 [24.25, 45] | 0.655 |
| Follow-up duration, months | 37 [23, 54] | 7 [4.25, 12.75] | <0.001 |
| SLE disease activity | | | |
|    Disease duration, months | 9 [1, 118.5] | 25 [2, 102.75] | 0.366 |
|    SLEDAI score | 12.5 [10.5, 16] | 10 [5.25, 16] | 0.060 |
|    PGA score | 2.0 [1.7, 2.3] | 1.85 [0.92, 2.3] | 0.099 |
|    Remission, *n* (%) | 0 | 5 (9.3) | 0.066 |
|    LLADS, *n* (%) | 1 (2.4) | 6 (11.1) | 0.216 |
|    Anti-dsDNA antibody positive, *n* (%) | 33 (78.6) | 34 (63.0) | 0.098 |
|    Low C3, *n* (%) | 41 (97.6) | 49 (90.7) | 0.339 |
|    C3 level, g/L | 0.46 ± 0.19 | 0.50 ± 0.21 | 0.321 |
|    Low C4, *n* (%) | 36 (87.5) | 41 (75.9) | 0.232 |
|    C4 level, g/L | 0.10 [0.04, 0.13] | 0.12 [0.06, 0.15] | 0.145 |
|    IgA, g/L | 2.44 [1.84, 3.70] | 2.1 [1.39, 2.84] | 0.133 |
|    IgM, g/L | 0.83 [0.54, 1.05] | 0.79 [0.49, 1.13] | 0.991 |
|    IgG, g/L | 11.4 [8.41, 15.5] | 10.04 [8.44, 12.95] | 0.247 |
| Percentage of CD19+ B cell | 21.0 [11.5, 24.9] | 11.2 [6.32, 19.68] | 0.139 |
| Renal manifestation | | | |
|    Serum creatinine, μmol/L | 63.70 [48.25, 88.07] | 65.00 [53.00, 77.25] | 0.248 |
|    Acute kidney injury, *n* (%) | 3 (7.1) | 6 (11.1) | 0.757 |
|    Chronic kidney disease, *n* (%) | | | |
|    Stage 1 | 26 (61.9) | 39 (72.2) | 0.284 |
|    Stage 2 | 12 (28.6) | 3 (5.6) | 0.005 |
|    Stage 3 | 3 (7.1) | 4 (7.4) | 1.000 |
|    Stage 4 | 1 (2.4) | 2 (3.7) | 1.000 |
|    Stage 5 | 0 | 6 (11.1) | 0.034 |
|    Proteinuria, g/d | 1.96 [0.64, 3.52] | 1.48 [0.18, 3.36] | 0.13 |
|    UACR, mg/mmol | 1559.06 [690.9, 2414.15] | 654.42 [29.52, 1943.18] | 0.010 |
|    Proteinuria, *n* (%) | 41 (97.6) | 40 (74.1) | 0.002 |
|    Hematuria, *n* (%) | 20 (47.6) | 32 (59.3) | 0.256 |
|    Leukocyturia, *n* (%) | 20 (47.6) | 22 (40.7) | 0.500 |
|    Cylindruria, *n* (%) | 4 (9.5) | 5 (9.3) | 1.000 |
| Histopathologic classification of lupus nephritis, *n* (%) | | | |
|    Pure II | 2 (4.8) | 0 | 0.189 |
|    Pure III | 3 (7.1) | 0 | 0.080 |
|    Pure IV | 12 (28.6) | 26 (48.1) | 0.052 |
|    Pure V | 7 (16.7) | 2 (3.7) | 0.070 |
|    III and V | 7 (16.7) | 5 (9.3) | 0.276 |
|    IV and V | 10 (23.8) | 14 (25.9) | 0.812 |

**Table 1** (*continued*)

| Characteristic | Conventional therapy (*n* = 42) | Belimumab (*n* = 54) | *P* value (Belimumab *vs* Conventional) |
|---|---|---|---|
| VI | 1 (2.4) | 0 | 0.437 |
| Unknown | 0 | 7 (13.0) | 0.017 |
| Concomitant treatment, *n* (%) | | | |
| Oral corticosteroid | 54 (100) | 54 (100) | |
| Antimalarials | 39 (92.9) | 48 (88.9) | 0.757 |
| Immunosuppressants | | | |
| Mycophenolate mofetil | 24 (57.1) | 30 (55.6) | 0.876 |
| Leflunomide | 1 (2.4) | 1 (1.9) | 1.000 |
| Tacrolimus | 3 (7.1) | 8 (14.8) | 0.397 |
| Azathioprine | 0 | 4 (7.4) | 0.129 |
| Rituximab | 0 | 1 (1.9) | 1.000 |
| Multi-target therapeutics | 7 (16.7) | 5 (9.3) | 0.276 |
| Steroid dose | | | |
| Steroid dose, mg/d | 40 (30, 50) | 30 (10, 50) | 0.054 |
| Daily dose >7.5 mg, *n* (%) | 40 (95.2) | 46 (85.2) | 0.207 |
| Daily dose >5 mg, *n* (%) | 40 (95.2) | 48 (88.9) | 0.457 |

Among the cohort, 92.7% (89/96) had renal biopsy-confirmed LN, with Class IV being the most common. As shown in Table 1, there was no significant difference in renal pathology types between the two groups. The SLEDAI score of the belimumab group was 10 [5.25,16], the median PGA score was 1.85 [0.92, 2.3], with a remission rate of 9.3% (5/54), and the median proteinuria level was 1.48 [0.18, 3.36] g/24 h, and the median serum creatinine level was 65.0 [53.0, 77.25] μmol/L, showing no significant difference from the conventional therapy group.. Notably, 4 (7.4%) of the patients in the belimumab group were undergoing renal replacement therapy, with one receiving peritoneal dialysis and three undergoing hemodialysis, whereas the conventional therapy group had none. Additionally, nine patients had an eGFR less than 30 mL/min/1.73 m$^2$, a category previously excluded from randomized controlled trials (*Furie et al., 2020*). In the belimumab group, anti-dsDNA antibodies were positive in 63.0% (34/54) of patients, 90.7% (49/54) had low C3 levels, and 75.9% (41/54) had low C4 levels. No significant differences were found between two groups.

Upon enrollment, all patients were on steroid therapy. As shown in Table 1, in the belimumab group, the median prednisone equivalent dose was 30 [10, 50] mg/day, while in the conventional therapy group, it was 40 mg/day [30, 50]. The difference was not statistically significant (*p* =0.054). In the belimumab group, 88.9% (48/54) of patients were treated with hydroxychloroquine, 55.6% (30/54) received mycophenolate mofetil, 14.8% (8/54) were given tacrolimus, and 9.3% (5/54) were administered multitarget therapeutics. No significant differences were found between the two groups.

## Renal manifestations

The majority of patients exhibited an improvement in renal function, as evidenced by evaluations of serum creatinine, hematuria, proteinuria, leukocyturia, cylindruria, and the UACR, which were conducted regularly throughout the study.

Notably, as shown in Table 2, in belimumab group, the UACR showed a significant reduction from 654.42 [29.52, 1943.18] mg/mmol at baseline to 210.00 [18.35, 961.52] mg/mmol by month 3 ($p = 0.008$), with a further decrease to 109.25 [22.60, 336.11] mg/mmol by month 6 ($p = 0.056$). Additionally, the proportion of patients presenting with proteinuria declined from 74.1% (40/54) at baseline to 64.8% (35/54, $p <0.001$) at month 3, 60.5% (26/43, $p <0.001$) at month 6, and 55.6% (10/18, $p = 0.023$) at month 12. Among the patients with hematuria at baseline (59.3%), the prevalence decreased to 37.0% (20/54), 27.9% (12/43), and 11.1% (2/18) by month 3, 6, and 12, respectively ($p = 0.008$, $p <0.001$, $p = 0.529$). Baseline assessments indicated that 53.7% (29/54) of the participants had already achieved either partial renal remission (PRR) or complete renal remission (CRR), as shown in Fig. 1. Their use of belimumab was intended to facilitate a reduction in corticosteroid utilization. By month 3, 27.8% (15/54) and 48.1% (26/43) attained CRR and PRR, respectively. Remarkably, by month 6, 95.3% (41/43) achieved either PRR or CRR, and by month 12, all 18 assessed patients (100%) reached either PRR or CRR. No renal flares were reported during the study.

In comparison, there were no significant differences in UACR, proteinuria, hematuria, leukocyturia, and serum creatinine at month 3, month 6, and month 12 in two groups (Table 3).

## Serologic features

As delineated in Table 2, the prevalence of positive anti-dsDNA antibodies among patients in belimumab group diminished over time: from 34 (62.96%) at baseline to 15 (27.77%, $p = 0.129$) at month 3, and further reducing to 14 (32.56%, $p = 0.015$) at month 6, as illustrated in Fig. 2A Additionally, a statistically significant reduction was observed in the percentage of $CD19^+$ B cells at 3, 6, and 12 months post-treatment, compared to baseline levels ($p <0.001$ at 3 and 6 months, $p = 0.013$ at 12 months), depicted in Fig. 2B. The number of patients with a low C3 level decreased from 49 (90.7%) at baseline to 44 (81.5%) at 3 months, 36 (83.7%) at 6 months, and 16 (88.9%) at 12 months ($p = 0.039$ at 3 months, $p = 0.302$ at 6 months, and $p = 0.111$ at 12 months). Similarly, the patient count with low C4 levels showed a reduction from 41 (75.9%) initially to 34 (63.0%) at 3 months, 27 (62.8%) at 6 months, and 12 (66.7%) at 12 months ($p <0.001$ at 3 and 6 months, $p = 0.025$ at 12 months). Figures 2C and 2D indicate that the levels of C3 and C4 after 3, 6, and 12 months of belimumab treatment were significantly elevated compared to baseline values.

In comparison, at month 6 and month 12, the C3 levels in the belimumab group were lower than those in the conventional therapy group (0.66 g/L vs 0.76 g/L, $p =0.02$), (0.60 g/L vs 0.75 g/L, $p =0.015$), as shown in Fig. 3A. Additionally, at month 12, the proportion of patients with low C3 level in the belimumab group was higher than those in the conventional therapy group (88.6% vs 61.9%, $p =0.037$). During the follow-up period,

**Table 2  Serologic and renal improvement in lupus nephritis with belimumab therapy at baseline and month 3, month 6, and month 12.**

| Category | Month 3 | Month 6 | Month 12 | Pvalue (Baseline vs Month 3) |
|---|---|---|---|---|
| **Renal manifestation** | | | | |
| Proteinuria, n (%) | 35 (64.8) | 26 (60.5) | 10 (55.6) | <0.001 |
| UACR, mg/mmol | 210.00 [18.35, 961.52] | 109.25 [22.60, 336.11] | 160.20 [94.61, 396.68] | 0.008 |
| Hematuria, n (%) | 20 (37.0) | 12 (27.9) | 2 (11.1) | 0.008 |
| Leukocyturia, n (%) | 6 (11.1) | 3 (7.0) | 1 (5.6) | 0.517 |
| Serum creatinine, μmol/L | 63.00 [51.00, 69.75] | 61.00 [52.00, 71.50] | 60.00 [51.00, 70.75] | 0.693 |
| **Serologic features** | | | | |
| Anti-dsDNA positive, n (%) | 15 (27.8) | 14 (32.6) | 9 (50) | 0.129 |
| Low C3, n (%) | 44 (81.5) | 36 (83.7) | 16 (88.9) | 0.039 |
| C3 level, g/L | 0.67 ± 0.14 | 0.66 ± 0.13 | 0.64 ± 0.15 | <0.001 |
| Low C4, n (%) | 34 (63.0) | 27 (62.8) | 12 (66.7) | <0.001 |
| C4 level, g/L | 0.15 [0.12, 0.19] | 0.16 [0.12, 0.19] | 0.14 [0.12, 0.16] | <0.001 |
| Percentage of $CD19^+$ B cell | 8.30 [4.85, 15.82] | 6.20 [3.80, 9.25] | 3.90 [2.12, 5.35] | <0.001 |
| IgA, g/L | 1.56 [1.21, 2.08] | 1.74 [1.40, 2.33] | 1.42 [1.06, 1.78] | <0.001 |
| IgM, g/L | 0.65 [0.40, 0.86] | 0.50 [0.47, 0.83] | 0.50 [0.36, 0.77] | <0.001 |
| IgG, g/L | 7.41 [5.69, 10.9] | 8.28 [6.65, 11.93] | 9.43 [7.25, 12.57] | <0.001 |
| **Steroid dose** | | | | |
| Steroid dose, mg/d | 17.5 [10, 25] | 10 [7.5, 15] | 10 [5, 10] | <0.001 |
| Daily dose >7.5 mg, n (%) | 45 (83.3) | 31 (72.1) | 10 (55.6) | <0.001 |
| Daily dose >5 mg, n (%) | 47 (87.0) | 34 (79.1) | 11 (61.1) | <0.001 |
| **Disease activity assessment** | | | | |
| SLEDAI score | 5 [2, 8] | 4 [2, 6] | 4 [2, 6] | <0.001 |
| PGA score | 0.95 [0.43, 1.5] | 0.8 [0.4, 1.0] | 0.8 [0.4, 1.1] | <0.001 |
| Remission, n (%) | 7 (13.0) | 9 (20.9) | 5 (27.8) | <0.001 |
| LLADS, n (%) | 9 (16.7) | 11 (25.6) | 7 (38.9) | <0.001 |

there were differences in the prevalence of positive anti-dsDNA antibodies, IgM levels, IgG levels, C4 levels, and the proportion of low C4 level, between the two groups, but these differences were not statistically significant.

## Steroid dose

At the commencement of the study, all 96 participants were prescribed steroids. In belimumab group, a significant reduction in the median steroid dose was observed: 30 [10, 50] mg/day at baseline to 17.5 [10, 25] mg/day by 3 months, 10 [7.5, 15] mg/day by 6 months, and 10 [5, 10] mg/day by 12 months ($p < 0.001$ at each time point). At baseline, 85.19% (46/54) of patients required steroid doses exceeding 7.5 mg/day. This proportion decreased to 83.33% (45/54, $p < 0.001$) by 3 months and further to 72.09% (31/43, $p < 0.001$) by 6 months. Similarly, the percentage of patients prescribed more than 5 mg of steroids daily decreased from 88.89% (48/54) initially to 87.04% (47/54, $p < 0.001$) by 3 months and 79.07% (34/43, $p < 0.001$) by 6 months, as illustrated in Figs. 4A and 4B.

As shown in Table 3 and Fig. 3B, in comparison, at month 3, the steroid dose in the belimumab group was significantly lower than in the conventional therapy group (17.5

Lin et al. (2024), *PeerJ*, DOI 10.7717/peerj.18028

**Table 3  Serologic and renal improvement at month 3, month 6, and month 12.**

| Category | Month 3 | | | Month 6 | | | Month 12 | | |
|---|---|---|---|---|---|---|---|---|---|
| | Conventional therapy | Belimumab | *P* value | Conventional therapy | Belimumab | *P* value | Conventional therapy | Belimumab | *P* value |
| Renal manifestation | | | | | | | | | |
| Proteinuria, *n* (%) | 29 (69.1) | 35 (64.8) | 0.663 | 22 (53.7) | 26 (60.5) | 0.529 | 22 (52.4) | 10 (55.6) | 0.821 |
| UACR, mg/mmol | 353.15 [50.95, 693.95] | 210.00 [18.35, 961.52] | 0.402 | 325.60 [50.45, 838.52] | 109.25 [22.60, 336.11] | 0.084 | 72.80 [44.35, 414.12] | 160.20 [94.61, 396.68] | 0.354 |
| Hematuria, *n* (%) | 14 (33.3) | 20 (37.0) | 0.707 | 17 (40.5) | 12 (27.9) | 0.222 | 9 (21.4) | 2 (11.1) | 0.56 |
| Leukocyturia, *n* (%) | 7 (16.7) | 6 (11.1) | 0.430 | 7 (16.7) | 3 (7.0) | 0.294 | 6 (14.3) | 1 (5.6) | 0.599 |
| Serum creatinine, $\mu$mol/L | 59.00 [53.50, 71.15] | 63.00 [51.00, 69.75] | 0.720 | 60.80 [51.50, 73.28] | 61.00 [52.00, 71.50] | 0.951 | 59.50 [53.20, 70.97] | 60.00 [51.00, 70.75] | 0.958 |
| Serologic features | | | | | | | | | |
| Anti-dsDNA positive, *n* (%) | 18 (42.9) | 15 (27.8) | 0.123 | 11 (26.2) | 14 (32.6) | 0.519 | 12 (28.6) | 9 (50) | 0.125 |
| Low C3, *n* (%) | 24 (57.1) | 38 (71.7) | 0.139 | 26 (61.9) | 34 (81.0) | 0.053 | 26 (61.9) | 16 (88.9) | 0.037 |
| C3 level, g/L | 0.72 [0.59, 0.84] | 0.66 [0.57, 0.79] | 0.109 | 0.76 [0.62, 0.82] | 0.66 [0.58, 0.73] | 0.02 | 0.75 [0.62, 0.81] | 0.60 [0.56, 0.72] | 0.015 |
| Low C4, *n* (%) | 26 (61.9) | 27 (50.9) | 0.285 | 19 (46.3) | 20 (48.8) | 0.825 | 22 (52.4) | 10 (62.5) | 0.489 |
| C4 level, g/L | 0.14 [0.12, 0.20] | 0.15 [0.12, 0.19] | 0.435 | 0.16 [0.12, 0.19] | 0.16 [0.12, 0.19] | 0.981 | 0.14 [0.12, 0.21] | 0.14 [0.12, 0.16] | 0.571 |
| IgA, g/L | 2.22 [0.58, 3.21] | 1.56 [1.21, 2.08] | 0.538 | 2.84 [1.46, 3.45] | 1.74 [1.40, 2.33] | 0.322 | 2.87 [2.59, 3.41] | 1.42 [1.06, 1.78] | 0.011 |
| IgM, g/L | 0.75 [0.54, 1.04] | 0.65 [0.40, 0.86] | 0.499 | 0.81 [0.65, 0.91] | 0.50 [0.47, 0.83] | 0.244 | 0.88 [0.59, 0.89] | 0.50 [0.36, 0.77] | 0.075 |
| IgG, g/L | 10.65 [9.56, 11.47] | 7.41 [5.69, 10.90] | 0.121 | 11.85 [10.62, 13.88] | 8.28 [6.65, 11.93] | 0.244 | 12.65 [9.85, 14.42] | 9.43 [7.25, 12.57] | 0.096 |
| Steroid dose | | | | | | | | | |
| Steroid dose, mg/d | 25 [20, 30] | 17.5 [10, 25] | 0.009 | 11.25 [10, 20] | 10 [7.5, 15] | 0.212 | 10 [5, 12.5] | 10 [5, 10] | 0.841 |
| Daily dos *e* > 7.5mg, *n* (%) | 40 (95.2) | 45 (83.3) | 0.135 | 35 (83.3) | 31 (72.1) | 0.214 | 23 (54.8) | 10 (55.6) | 0.595 |
| Daily dose >5mg, *n* (%) | 40 (95.2) | 47 (87.0) | 0.310 | 36 (85.7) | 34 (79.1) | 0.422 | 25 (59.5) | 11 (61.1) | 0.517 |

Lin et al. (2024), *PeerJ*, DOI 10.7717/peerj.18028

Peerj

**Table 3** (*continued*)

| Category | Month 3 | | | Month 6 | | | Month 12 | | |
|---|---|---|---|---|---|---|---|---|---|
| | Conventional therapy | Belimumab | *P* value | Conventional therapy | Belimumab | *P* value | Conventional therapy | Belimumab | *P* value |
| Disease activity assessment | | | | | | | | | |
| SLEDAI score | 6 [4, 8] | 5 [2, 8] | 0.067 | 6 [4, 8] | 4 [2, 6] | 0.014 | 4 [2.5, 7.5] | 4 [2, 6] | 0.444 |
| PGA score | 1.0 [0.9, 1.7] | 0.95 [0.43, 1.5] | 0.026 | 1.0 [0.9, 1.5] | 0.8 [0.4, 1.0] | 0.004 | 0.95 [0.4, 1.08] | 0.8 [0.4, 1.1] | 0.516 |
| Remission, *n* (%) | 1 (2.4) | 7 (13.0) | 0.137 | 2 (4.8) | 9 (20.9) | 0.026 | 5 (11.9) | 5 (27.8) | 0.257 |
| LLADS, *n* (%) | 2 (4.8) | 9 (16.7) | 0.135 | 4 (9.5) | 11 (25.6) | 0.052 | 15 (35.7) | 7 (38.9) | 0.815 |

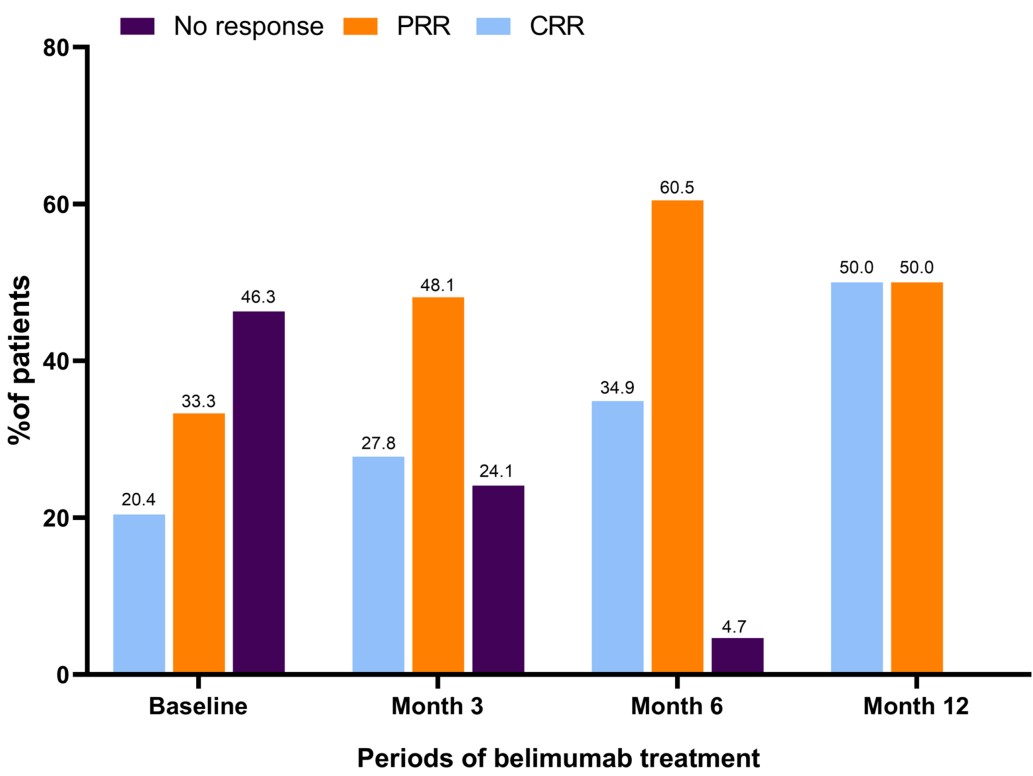

**Figure 1   Renal response during belimumab treatment.** At baseline, 29 patients (53.7%) achieved CRR or PRR. At month 3, 15 (27.8%) and 26 (48.1%) patients achieved CRR and PRR. At month 6, 15 (34.9%) and 26 (60.5%) patients achieved CRR and PRR. At month 12, 9 (50.0%) and 9 (50.0%) patients achieved CRR and PRR. CRR, complete renal response; PRR, partial renal response.

mg/d *vs* 25.0 mg/d, $p = 0.009$). At month 6 and month 12, the steroid doses in the belimumab group were lower than in the conventional therapy group, but the differences were not statistically significant. During the follow-up period, the proportion of patients in the conventional therapy group requiring more than 7.5 mg or 5 mg of steroids daily decreased, with no statistically significant differences when compared to the belimumab group.

### Assessment of overall clinical response and disease activity

In belimumab group, the median SLEDAI score exhibited a significant reduction from an initial value of 10 [5.25, 16] to 5 [2, 8] at 3 months, 4 [2, 6] at 6 months, and maintained at 4 [2, 6] at 12 months ($p < 0.001$ at each time point). Prior to belimumab treatment, 23 patients (42.6%) were classified within the severe category, while 14 patients (25.9%) were deemed to have moderate disease severity. Following the treatment with belimumab, there was a significant reduction in the proportion of patients classified as severe, with the majority showing improvements in SLEDAI scores, as shown in Fig. 4C. At the commencement of belimumab therapy, remission was observed in 9.3% (5/54) of the patients. Subsequent evaluations revealed an increase in the number of patients in remission to 7 (13.0%, $p < 0.001$) at 3 months, 9 (20.9%, $p < 0.001$) at 6 months, and 5 (27.8%) at 12 months.

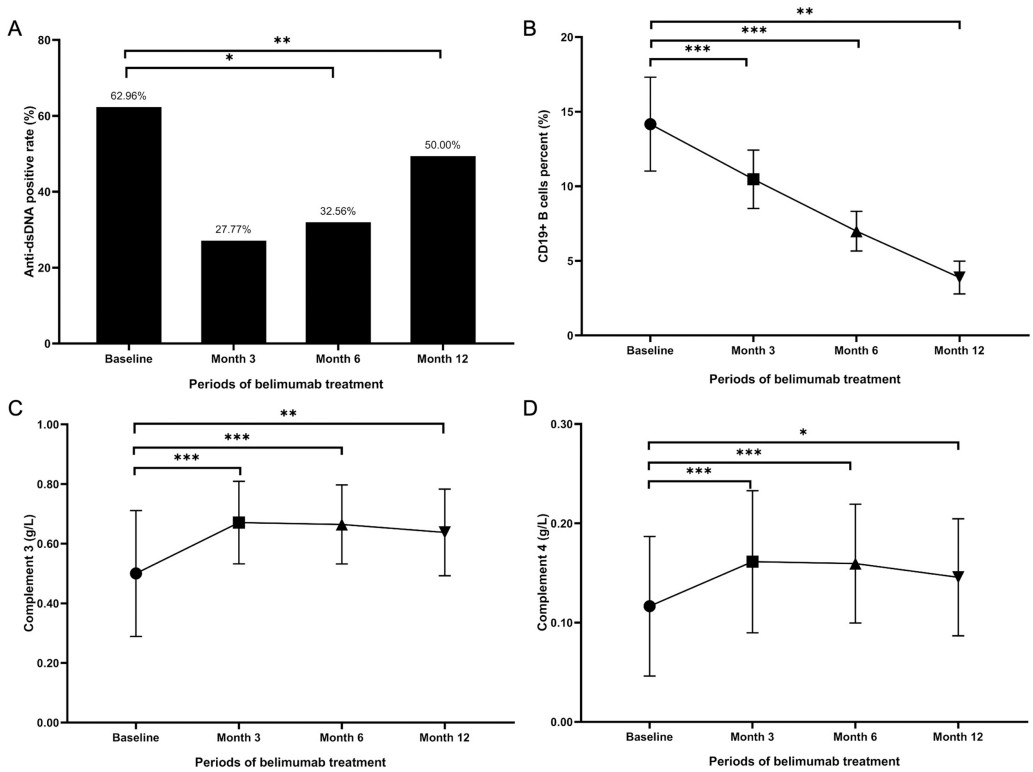

**Figure 2** **Serological changes in LN patients treated with belimumab.** (A) Anti-dsDNA antibody positive rate (%). (B) Percentage of CD19+ B cells (%). (C) Complement 3 level (g/L). (D) Complement 4 level (g/L). * $p < 0.05$; ** $p < 0.01$; *** $p < 0.001$.

Concurrently, the number of patients achieving LLDAS escalated from 6 (11.10%) at baseline to 9 (16.67%, $p < 0.001$) at 3 months, 11 (25.58%, $p = 0.003$) at 6 months, and 7 (38.89%, $p = 0.389$) at 12 months, as shown in Fig. 4D.

In comparison, at month 6, the SLEDAI scores of patients in the belimumab group were significantly lower than those in the conventional therapy group (4 *vs* 6, $p = 0.014$), and the number of patients achieving remission in the belimumab group was significantly higher than in the conventional therapy group (9 *vs* 2, $p = 0.026$), as depicted in Table 3, Figs. 3C and 3D. Additionally, at month 3 and month 6, the PGA scores of patients in the belimumab group were significantly lower than those in the conventional group,(0.95 *vs* 1.0, $p = 0.026$), (0.8 *vs* 1.0, $p = 0.004$).

## Treatment-emergent adverse events

Throughout the entire follow-up period, as detailed in Table 4, the majority of treatment-emergent adverse events (TEAEs) were infections of a mild nature, comprising 13 cases of urinary tract infection, 14 cases of respiratory tract infection, three cases of herpes zoster, and one case of gastrointestinal infection. Moreover, six patients reported nervous system disorders, two cases occurred in psychiatric disorders, and another associated with musculoskeletal discomfort. The TEAEs were predominantly mild to moderate in severity

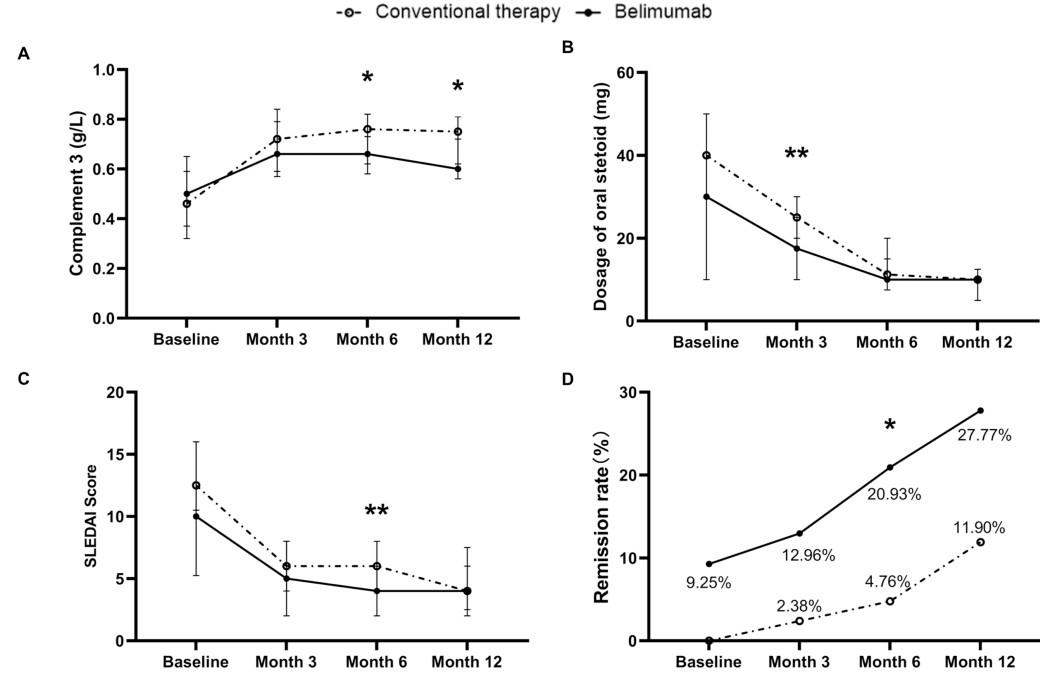

**Figure 3** **Comparison of efficacy between conventional therapy and belimumab groups.** (A) Complement 3 level (g/L). (B) Dosage of oral steroid (mg/d). (C) SLEDAI score. (D) Remission rate (%). * $p <$ 0.05; ** $p < 0.01$; *** $p < 0.001$.

**Table 4** **Treatment-emergent adverse events (TEAEs) that occurred during the period of belimumab therapy and conventional therapy.**

| TEAEs, n (%) | Conventional therapy | Belimumab | P value |
|---|---|---|---|
| Infusion reaction | 0 | 0 | |
| Allergic reaction | 0 | 0 | |
| Infection | | | |
|     Respiratory tract infection | 10 (23.8) | 4 (7.4) | 0.049 |
|     Urinary tract infection | 7 (16.7) | 6 (11.1) | 0.695 |
|     Herpes simplex | 0 | 0 | |
|     Herpes zoster | 2 (4.8) | 1 (1.9) | 0.825 |
|     Gastrointestinal infection | 0 | 1 (1.9) | 1.000 |
| Leukopenia | 0 | 0 | |
| Gastrointestinal disorders | 0 | 0 | |
| Psychiatric disorders | 1 (2.4) | 1 (1.9) | 1.000 |
| Nervous system disorders | 4 (9.5) | 2 (3.7) | 0.457 |
| Musculoskeletal disorders | 0 | 1 (1.9) | 1.000 |
| Total | 22 (52.4) | 16 (29.6) | 0.024 |

and were ameliorated through symptomatic management. Throughout the study, no instances of mortality were observed.

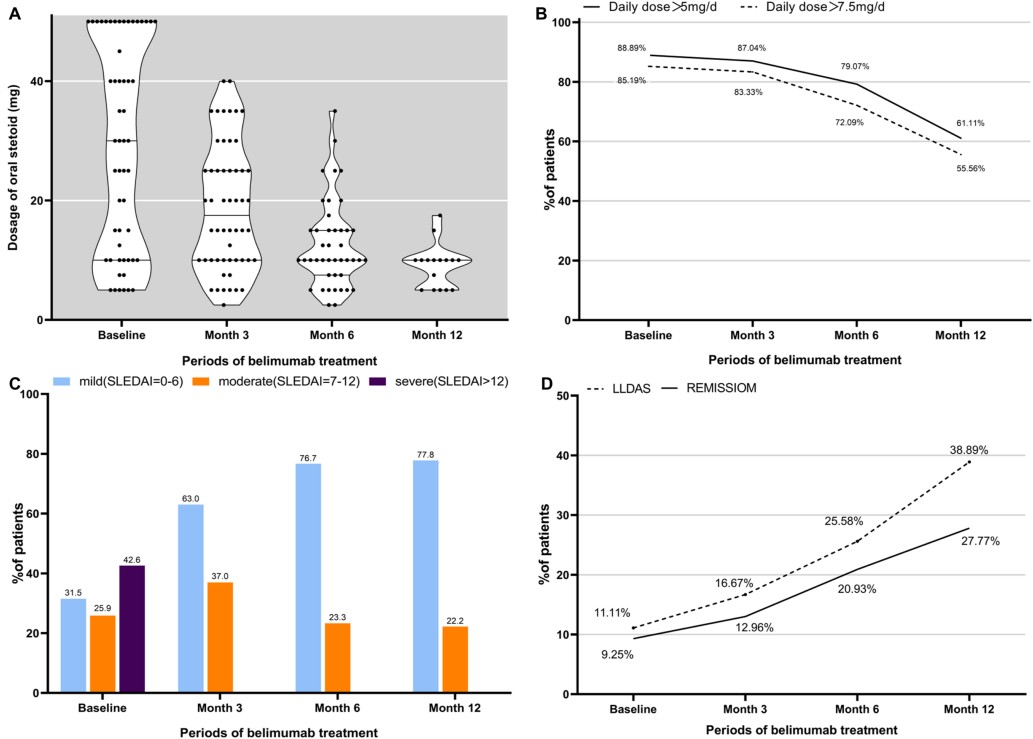

**Figure 4 Steroid dose (prednisone equivalent) and overall disease assessment during belimumab treatment.** (A) The oral prednisone equivalent daily dose of glucocorticoids significantly decreased from baseline to month 3, month 6, and month 12 ($p < 0.001$). (B) The proportion of patients taking steroids above 5 mg or 7 mg per day had significantly decreased. (C) Disease severity subgroups based on SLEDAI score in the cohort. (D) The treatment goals achieved in our cohort. LLDAS, Lupus Low Disease Activity State.

Notably, the incidence of TEAEs in the belimumab group was significantly lower than in the conventional therapy group (29.6% *vs* 52.4%, $p = 0.024$). Particularly significant was the reduced incidence of respiratory tract infections in the belimumab group, which was notably lower (7.4% *vs* 23.8%, $p = 0.049$).

## DISCUSSION

Despite significant therapeutic advancements, some LN patients fail to achieve CRR, leading to a poor long-term prognosis. Moreover, even for those who do achieve CRR, long-term therapy with steroids and immunosuppressants can result in numerous treatment-related adverse effects, some of which may be life-threatening (*Ji et al., 2022*). Belimumab, a promising biologic agent, is recommended for patients with inadequate response to standard treatment, with the goal of achieving renal remission, reducing flare and decreasing steroid dosage (*Furie et al., 2020*; *Fanouriakis et al., 2019*). However, in clinical practice, meeting the eligibility criteria of clinical trials can be challenging for patients. For instance, the efficacy of belimumab in patients undergoing renal replacement therapy remains unclear. Therefore, more real-world studies are warranted. This study analyzed the

baseline characteristics and outcomes of LN patients receiving conventional therapy and add-on therapy with belimumab, with the aim of evaluating and comparing the real-world efficacy and tolerability of add-on therapy with belimumab. The adjunctive therapy with belimumab conferred benefits in terms of renal response and a reduction in steroid dosage. Additionally, our research demonstrated the safety of belimumab in treating LN.

In our study, both groups exhibited a significant reduction in the proportion of patients experiencing hematuria, cylindruria, proteinuria, and leukocyturia. Additionally, renal function was noted to remain stable during the follow-up period. Consistent with findings from previous studies, the addition of belimumab to standard treatment for LN patients resulted in an overall improvement in clinical response and disease activity (*Tan et al., 2023*; *Yu et al., 2023*; *Navarra et al., 2011*; *De Oliveira et al., 2020*). Serological parameters indicated that patients with abnormal levels at baseline showed improvement towards normal levels, which persisted throughout the treatment process, including anti-dsDNA antibodies, C3, and C4.

Notably, approximately ninety-five percent of our patients who received add-on therapy with belimumab achieved partial or complete renal response at their last visit, accompanied by a significant increase in the proportion of patients achieving LLDAS and remission. It is noteworthy that all six LN patients who received belimumab in addition to conventional therapy presenting with acute kidney injury (AKI) experienced varying degrees of recovery in renal function. However, among the three patients with AKI who received conventional therapy only, one patient's kidney function never recovered during the follow-up period and eventually started peritoneal dialysis. Furthermore, compared with conventional therapy, belimumab therapy led to a more rapid reduction in the daily dosage of steroids. Additionally, most patients with normal serological markers did not experience deterioration, highlighting the crucial role of belimumab in preventing disease flare, a finding consistent with other observational studies (*Sishi et al., 2023*; *Tan et al., 2023*). Indeed, no renal flares were observed in our cohort.

As a complement to randomized controlled trials, our study encompassed a broader spectrum of LN patients, including those at stages 4 and 5 chronic kidney disease, individuals undergoing hemodialysis or peritoneal dialysis, and patients who had previously received rituximab before initiating belimumab treatment. Belimumab appears to offer additional benefits for LN patients undergoing kidney replacement therapy. This finding is consistent with the study by *Liu et al. (2022)* which reported that belimumab, when combined with conventional therapy, improved serological features and reduced disease activity in LN patients undergoing dialysis. Similarly, *Binda et al. (2020)* demonstrated that adjunctive therapy with belimumab could prevent immunologic flare in LN patients receiving peritoneal dialysis or following kidney transplantation. Our study corroborated these findings. In our belimumab group, three patients were undergoing hemodialysis, and one patient was undergoing peritoneal dialysis. Following belimumab treatment, these patients exhibited increases in complement levels, seroconversion of anti-dsDNA antibodies, and reductions in steroid dosage.

Our study also demonstrated that belimumab exerts a glucocorticoid-reducing effect in patients with LN, consistent with findings from previous studies on SLE patients. This effect

can lead to a reduction in the duration and average dose of steroid treatment. Specifically, at month 3, the steroid dosage was decreased in 36 out of 54 patients receiving add-on therapy with belimumab. However, it is notable that none of the patients completely discontinued steroids in our study, regardless of whether belimumab was added to the conventional therapy, possibly due to our conservative treatment approach. In contrast, a study from Italy reported that 35% of their patients receiving belimumab as adjunctive therapy discontinued steroids indefinitely, while the steroid dosage in other patients was reduced to approximately 40% of the baseline (*Binda et al., 2020*). Similarly, a real-world report by *Tan et al. (2023)* demonstrated that 8.0% of patients receiving belimumab treatment discontinued steroids altogether, while 77.0% experienced a reduction in dosage. *Sishi et al. (2023)* conducted a retrospective analysis involving 112 patients receiving belimumab in combination with conventional therapy, compared to 112 matched control patients receiving conventional therapy only. They found that combination therapy with belimumab effectively reduced steroid dosage, although no significant difference was observed between the two groups (*Sishi et al., 2023*). These findings align with the results of our study, where combination therapy with belimumab effectively reduced the steroid dosage.

Furthermore, in line with previous studies, our findings also indicate that patients generally exhibit good tolerance to belimumab (*Sishi et al., 2023*; *Tan et al., 2023*; *Liu et al., 2022*; *Shipa et al., 2021*). The most common TEAEs observed in our belimumab group were infections, primarily urinary tract infections. It is important to note that these infections cannot be solely attributed to belimumab, as patients were concurrently receiving steroids and immunosuppressants, which can also increase susceptibility to infections. Our study revealed that, compared to conventional therapy, patients who received a combination therapy with belimumab experienced a lower incidence of TAEAs. This reduction may be attributed to a more rapid tapering of steroids and immunosuppressants. Moreover, the infections reported were predominantly mild in nature, and no patient discontinued belimumab treatment due to these events. *Liu et al. (2022)* suggested that belimumab use in dialysis patients is safe, reporting only one case of pulmonary infection among seven dialysis patients in their study. In our study, one out of four dialysis patients experienced gastrointestinal infection in our cohort. Therefore, larger-scale studies involving dialysis patients are warranted to comprehensively evaluate the safety profile of belimumab use in this population.

Our study has several limitations that need to be acknowledged. These include the relatively short follow-up duration, the single-center design, the small sample size. Additionally, in our cohort, some patients were clinically diagnosed with LN and lacked renal pathological data, which means the diagnosis of LN could not be fully confirmed and the belimumab group notably lacked cases of Classes II, III, and VI lupus nephritis. Therefore, our data may have a certain degree of selection bias and may not represent all patients with LN. However, given the recent approval of belimumab for treating SLE in China, our study contributes valuable objective data to the research on the safety and efficacy of belimumab specifically in Chinese patients with LN. To address these limitations and provide more robust evidence, future studies should aim for a multi-center design with a larger sample size and a longer follow-up duration. This would allow for a more

comprehensive evaluation of the therapeutic effects and potential adverse events associated with belimumab therapy in this particular population.

## CONCLUSION

Our study provides compelling evidence supporting the beneficial effects of belimumab on renal performance improvement, disease activity reduction, glucocorticoid dose reduction, and the diminution of TEAEs in Chinese patients with LN. Nevertheless, further controlled, large-scale, randomized clinical trials are warranted to confirm and validate these findings. Conducting such trials would enhance the strength of evidence regarding the efficacy and safety of belimumab as a therapeutic option for LN patients, ultimately guiding clinical practice and improving patient outcomes.

### Funding

This work was supported by National Natural Science Foundation of China (No. 82300803), Fujian Research and Training Grants for Young and Middle-aged Leaders in Healthcare (2022QNRCYX-XYF), and the Outstanding Young Talents Program of the First Affiliated Hospital of Fujian Medical University (YJCQN-A-XYF2021). The funders had no role in study design, data collection and analysis, decision to publish, or preparation of the manuscript.

### Grant Disclosures

The following grant information was disclosed by the authors:
National Natural Science Foundation of China: 82300803.
Young and Middle-aged Leaders in Healthcare: 2022QNRCYX-XYF.
Outstanding Young Talents Program of the First Affiliated Hospital of Fujian Medical University: YJCQN-A-XYF2021.

### Competing Interests

The authors declare there are no competing interests.

### Author Contributions

- Zishan Lin conceived and designed the experiments, performed the experiments, analyzed the data, prepared figures and/or tables, authored or reviewed drafts of the article, and approved the final draft.
- Bingjing Jiang performed the experiments, analyzed the data, prepared figures and/or tables, authored or reviewed drafts of the article, and approved the final draft.
- Wenfeng Wang analyzed the data, prepared figures and/or tables, and approved the final draft.
- Caiming Chen analyzed the data, prepared figures and/or tables, and approved the final draft.

- Yujia Wang analyzed the data, prepared figures and/or tables, and approved the final draft.
- Jianxin Wan analyzed the data, prepared figures and/or tables, and approved the final draft.
- Yanfang Xu conceived and designed the experiments, authored or reviewed drafts of the article, and approved the final draft.

## Human Ethics

The following information was supplied relating to ethical approvals (i.e., approving body and any reference numbers):

This study was performed in accordance with the Helsinki Declaration and approved by the ethics committee of the First Affiliated Hospital of Fujian Medical University [2015]084-2. Written informed consent was obtained from each patient, and data are available upon request.

## Data Availability

The raw data is available in the Supplementary Files.

## Supplemental Information

Supplemental information for this article can be found online at http://dx.doi.org/10.7717/peerj.18028#supplemental-information.

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
