# Peer review of "Clinical outcomes in lupus nephritis patients treated with belimumab in real-life setting: a retrospective comparative study in China"

_PeerJ, doi:10.7717/peerj.18028_

## Round 0.1 · original submission · Major Revisions

· Academic Editor

Major Revisions

Dear Dr. Lin and Dr. Xu,

If you feel you can revise your manuscript according to the reviewers' comments, please revise your manuscript and submit it. Please also send us your written responses to each of the reviewers' comments.

Yours,

Yoshi

Prof. Yoshinori Marunaka, M.D., Ph.D.

Reviewer 1 ·

Basic reporting

To the Authors,

Thank you for the opportunity to review the manuscript.

In this paper, the authors investigated the clinical outcomes in lupus nephritis patients treated with belimumab in real-life setting. And the authors showed belimumab improved renal and serological parameters, reduce disease activity, and lessen corticosteroid dependence. This topic is interesting; however, several problems should be resolved prior to acceptance for publication in Peer J, as shown below.

<Specific Comments>
Firstly, this study demonstrates that belimumab is effective for lupus nephritis (LN) and allows for a reduction in glucocorticoids without recurrence of LN. However, this study is single-arm, and it is unclear whether it is more effective compared to conventional treatment. In other words, it is unclear how this study presents novelty compared to previously reported studies, as introduced by the authors. For example, presenting a comparison with conventional treatment using historical controls would provide more convincing data. As it stands, the current data alone do not provide sufficient novelty for publication. The authors should clarify the strengths of this study more explicitly.

The authors diagnose LN based on renal biopsy findings or urinalysis. Therefore, the LN class is unknown in some patients. Is there a possibility that patients without renal pathology findings might have LN class I or II? The lack of pathological findings is considered a significant limitation.

Experimental design

no commen

Validity of the findings

no commen

Additional comments

no commen

Reviewer 2 ·

Basic reporting

no comment

Experimental design

no comment

Validity of the findings

no comment

Additional comments

This article is an observational study of the use of belimumab in patients with lupus nephritis. The study has a small sample size, a relatively simple design, and lacks a control group, making it difficult to provide new high-quality evidence for the treatment of LN. In addition, large-sample RCTs have been published on the use of belimumab in LN, so the innovation of this article is limited.
In terms of the choice of statistical methods, I think there are some problems in this article, mainly the following points:

1. Line 128 – 129: “Continuous variables were presented as mean ± standard deviation (SD) or median with range …” Please explain what kind of continuous variables were presented as means / medians. If the presentation format was selected based on the distribution characteristics of the variable, please explain the test method used to determine whether the variable is normally distributed.
2. Line 129 – 130: “… and compared using the t-test or one-way analysis of variance (ANOVA)” Both t-test and ANOVA require that the variables be normally distributed. According to the results of this manuscript, this should be paired t test and paired sample Wilcoxon signed rank test?

---

## Round 0.2 · accepted · Accept

· Academic Editor

Accept

Congratulations again, and thank you for your submission.

Best regards,

Yoshi

Prof. Yoshinori Marunaka, M.D., Ph.D.

Reviewer 1 ·

Basic reporting

To the authors
Thank you for the opportunity to review the revised manuscript. The comments raised by the reviewer are well addressed.

Experimental design

Please, see the comment for the Basic reporting.

Validity of the findings

Please, see the comment for the Basic reporting.

Additional comments

Please, see the comment for the Basic reporting.